# A Systematic Review of the Aerobic Exercise Program Variables for Patients with Non-Specific Neck Pain: Effectiveness and Clinical Applications

**DOI:** 10.3390/healthcare11030339

**Published:** 2023-01-24

**Authors:** Eleftherios Paraskevopoulos, George A. Koumantakis, Maria Papandreou

**Affiliations:** 1Department of Physiotherapy, University of West Attica, 12243 Athens, Greece; 2Laboratory of Advanced Physiotherapy, University of West Attica, 12243 Athens, Greece

**Keywords:** aerobic exercise, non-specific neck pain, pain, function, disability

## Abstract

Studies have shown that aerobic exercise (AE) may improve symptoms related to non-specific neck pain (NNP); however, the variables of the exercise programs and the overall effectiveness of AE have not been evaluated in a systematic review. Therefore, this review aimed to describe and discuss the variables of the AE programs used in clinical trials for patients with NNP. Included studies were analyzed for the selected AE variables such as intensity, frequency, duration, delivery, supervision, and adherence. The PEDro scale was used to assess the methodological quality of the studies. From the literature search, six studies met the inclusion criteria and were evaluated. After reviewing all the included studies, it was found that a range of AE interventions were used such as cycling, brisk walking, aerobics, stationary bike, treadmill running, circuit training, and swimming. Further, the duration was between 30 and 45 min for each session, with or without progressive increases from week to week. The intervention periods ranged from 1 month to 6 months in duration. Most studies used AE three times per week. Furthermore, exercise intensity was measured with either subjective (BORG) or objective measures (heartrate reserve). Justification for the specified intensity and reporting of adverse events was reported only in two studies and differed between studies. Exercise interventions were poorly reported. This review showed that moderate-intensity AE undertaken three times per week, in patients with NNP, may be beneficial for pain and function; however, the development of reporting standards is essential for the successful replication of studies.

## 1. Introduction

Neck pain affects approximately 70% of the global population, with women being more at risk than men [1,2]. Of this percentage of patients, 50 to 85% will experience symptoms of neck pain again within 5 years after the first onset [2]. Neck pain constitutes a significant personal burden and can also seriously affect the healthcare system and economic structure of a country [3,4]. Furthermore, research has shown that the prevalence of neck pain will likely increase more in the future with the aging population [5]. Most patients with neck pain are usually classified as suffering from “non-specific” neck disorders [6]. Non-specific neck pain (NNP) refers to pain in the cervical region that occurs without trauma, symptoms of structural pathology, or neurological symptomatology.

Studies and one expert consensus suggest that exercise remains the mainstay treatment for the management of NNP [7,8,9]. A recent systematic review that examined the effectiveness of different exercise interventions, including motor control exercises, strengthening exercises, yoga, or multimodal exercise interventions for NNP, concluded that although exercise was effective, there was not one superior exercise intervention over the other [10].

It is important to note that aerobic exercise has been shown to be an effective intervention for the management of many functional disorders such as low back pain [11], fibromyalgia [12], myofascial pain [13], and pain perception [14]. Furthermore, recent research has highlighted the benefits of aerobic exercise on physical and mental outcomes that may improve the overall quality of life [15,16]. Until today, only one review has commented on the beneficial effects of exercise in NNP, including aerobic exercise [17]. However, more studies have been published since then [9,18,19,20]. Furthermore, a systematic review examining the clinical significance of the published studies has not been conducted yet to evaluate the clinical significance of aerobic exercise in patients with NNP. Based on promising results from recent studies and the lack of systematic reviews examining the clinical effectiveness of aerobic exercise for NNP management, it seems important to evaluate the findings of the available studies and investigate the exact exercise program variables such as frequency, intensity, duration, and type that result in positive outcomes based on the FITT principle (frequency, intensity, time, and type) [21].

Thus, this review aimed to assess the effectiveness of aerobic exercise in NNP and more specifically the exercise program parameters that were employed in the eligible studies. We anticipated that an updated systematic review would provide more definitive answers regarding the effectiveness of aerobic exercise in NNP in several outcomes regularly used in this clinical population.

## 2. Materials and Methods

This systematic review was conducted according to the Preferred Reporting Items for Systematic Reviews and Meta-Analyses (PRISMA) statement. This study was prospectively registered on PROSPERO (CRD42022373528).

### 2.1. Search Methods

International electronic databases were used for the literature search, including PubMed, Scopus, and PEDro, from inception up to October 2022. We consulted with experts in the field, manually reviewed the reference lists of articles that fulfilled the eligibility criteria, and searched the grey literature for eligible articles. The keywords that were used for the search were neck pain, cervical spine, aerobic exercise, running, walking, cycling, dancing, and a similar combination of words. Combining keywords in relation to the populations (NNP, neck pain, chronic neck pain) and the intervention (aerobic exercise, aerobics, jogging, cycling, walking, swimming, etc.) using Boolean operators allowed us to construct different research equations, introduced into the databases. We consulted the Cochrane Back and Neck Review Group for our search strategy. All studies were downloaded into Endnote X8 for screening.

### 2.2. Study Selection

Randomized controlled trials (RCTs) published in English were screened for eligibility by title and abstract and then by full text. Two independent reviewers (EP, MP) performed the search, as well as the full inclusion process using the PICOS framework (P = participants; I = interventions; C = comparison; O = outcomes, S = study design). The full texts of all potentially eligible studies were retrieved for the last process of the evaluation. The process of selecting the final eligible studies was performed by consensus. Disagreements were resolved by a third reviewer (GK) when required for the final judgment.

### 2.3. Eligibility Criteria Based on the PICOS Framework

#### 2.3.1. Participants

Studies were considered if they included adult patients (≥18 years) of either sex. Only studies that recruited symptomatic participants with a chief complaint of musculoskeletal neck pain were included in this review. Studies were eligible if they recruited patients with a duration of symptoms of at least 4 weeks, as previously suggested [13]. Studies that recruited patients with a history of traumatic injury, surgery or systemic diseases, or diseases related to other areas, such as the shoulder, were excluded. Furthermore, studies that recruited patients with neuropathies/radiculopathies (clinically tested by positive Spurling, cervical traction, and brachial plexus tests) were also excluded.

#### 2.3.2. Intervention

Studies that evaluated the effectiveness of aerobic exercise were eligible. We considered studies that delivered aerobic exercise programs with either light, moderate, or high-intensity aerobic exercise. Moreover, studies that evaluated the effectiveness of aerobic exercise with or without other interventions were eligible.

#### 2.3.3. Comparison Groups

Studies were considered eligible for inclusion if they compared groups of patients that received any other interventions or placebo or sham treatment.

#### 2.3.4. Outcome Measures

Studies were included in this review if they analyzed at least one of the following outcome measures at baseline and final follow-up assessment: pain that was measured with a subjective tool (i.e., Visual Analogue Scale or Numerical Rating Scale), disability and quality of life with a questionnaire (i.e., Neck Disability Index and Short-Form-36 Health Survey (SF-36), respectively). Further, studies were eligible if they reported their exercise program parameters such as intensity, frequency, duration, delivery, supervision, and adherence.

#### 2.3.5. Study Design

Only RCTs were considered eligible for this review. Case studies and pilot studies were excluded. Articles in English and Greek were accepted for inclusion.

### 2.4. Methodological Quality Assessment

The studies that were included in this review were classified for their methodological quality and risk of bias according to PEDro. This is a tool that consists of 11 items related to the validity of the articles assessed and is considered highly reliable [18]. The PEDro final score ranges from 0 (low quality) to 10 (high quality), with each criterion contributing 1 point. If a criterion is not described or was unclear, no point is awarded. The first criterion is related to external validity; however, this is not included in the final score. The remaining 10 items concern internal validity [22]. These items are used to assess methodological issues related to random allocation, allocation concealment, baseline comparability, blinding of therapists, patients, and raters, experimental mortality, intention-to-treat analysis, statistical comparisons and point measures, and measures of variability. Items 2–9 may identify studies that are likely to be internally valid, while items 10–11 may identify studies that provide sufficient statistical information and make their findings interpretable.

Two of the authors of the review (EP, MP) assessed all studies, and the third author (GK) was available in case of disagreement if needed. Based on the results of PEDro, the methodological quality of each study can be considered high (≥7), moderate (5 or 6), or poor (≤4). For the methodological assessment, we measured the interrater agreement with the intraclass correlation coefficient (ICC) using the SPSS software (IBM version 28).

### 2.5. Data Extraction

Two of the authors (EP, MP) independently extracted all data using a standardized form and collected information related to participant characteristics, study design, follow-up, interventions (type, duration, and the number of sessions), comparison groups characteristics, outcomes, the intensity of the program, and availability of supervision.

### 2.6. Evidence Synthesis

A narrative synthesis was used to synthesize the data of the included studies and comment on the results. The three phases of narrative synthesis included “developing a preliminary synthesis, exploring relationships within and between studies, and determining the robustness of the synthesis” [23]. The data of the included studies were described qualitatively, and the results were evaluated by the authors [24].

## 3. Results

We identified 608 trials after removal of duplicates that were potentially relevant and after reading the titles and abstracts, 9 articles were found as potentially eligible for review. Full texts of the nine articles were scrutinized for eligibility based on our inclusion and exclusion criteria, and three articles were excluded. The reasons for the removal of the studies can be found in Figure 1.

### 3.1. Participants

The characteristics of the participants from the included studies are available in Table 1. The final sample in our qualitative synthesis was 549 participants after 64 dropouts (11.6%). The vast majority of the studies included samples with mean ages that ranged from 36 to 55 years old. Further, 439 subjects from the whole sample were women. In some studies, participants were recruited after medical referral, except in two studies [25,26]. Moreover, another two studies [9,18] did not recruit participants only after medical referral but also through advertisement. The duration of symptoms varied between participants from 4 weeks to at least 6 months during recruitment, while some of the studies did not mention the symptom duration of the participants [9,18,26]. 

### 3.2. Interventions

Characteristics of the interventions are listed in Table 1. Aerobic exercise was beneficial for patients with NNP. Aerobic exercise was applied alone [9,18,26] or with other interventions [19,20,25]. The aerobic exercise included cycling [9,18,19,25], brisk walking [9,18], aerobics [26], biking on a stationary bike [26], treadmill running [26], circuit training [26], and swimming [25]. Duration of aerobic exercise was 30 [25,26] or 45 [9,18,19,20] min with a progressive increase [19,20,25] or without a progressive increase [9,18,26] to the aforementioned durations. For the intensity of the exercise program, in studies that evaluated intensity based on subjective measures such as the BORG scale, the intensity was estimated as moderate to high for two studies [9,18] and moderate for one study [25]. Exercise intensity based on the rate of perceived exertion using the BORG scale was estimated as moderate or high after considering the recommendations of a previous study [27]. Three studies evaluated exercise intensity using objective measures such as the heartrate reserve (HRR-67%) [26] and maximum heartrate (MHR-60% of maximum) [19,20]. Based on these percentages of HRR and MHR, we determined intensity as moderate for the two studies that used MHR and high for the study that used the HRR, following suggestions from previous research [28]. Aerobic exercise with or without other interventions was compared to either strengthening exercises of the neck [19,20], a control group [9,18], lectures on health promotion [26], or pain education [25].

### 3.3. Outcome Measures

Pain was assessed in five studies [9,18,20,25,26], whereas function and disability were assessed in all studies except from one [18], but with several different outcome measures, such as the global perceived effect (GPE) [25], level of physical activity, psychological well-being, and perceived ability to perform daily activities [9], Neck Disability Index (NDI) [20], Fear-Avoidance Beliefs Questionnaire (FABQ) [20] and Work Ability Index (WAI) [19]. Due to the large heterogeneity of the outcome measures that were used, the generation of a pooled estimate was not feasible.

### 3.4. Methodological Quality

The final score of all studies ranged from 5 to 8, as shown in Table 2. According to the criteria provided in the PEDro, three studies were classified as high methodological quality RCTs. Two criteria that were related to patient and therapist blinding were not met in any of the studies. Further, only two studies blinded the assessor [20,25], and only two studies provided adequate follow-up assessment [19,20]. Reviewers’ agreement was high when assessing the methodological quality of the studies (ICC = 0.90).

### 3.5. Program Variable Analysis

Exercise frequency and intensity were reported in all studies. Exercise intensity was specified in all studies; however, justification for the specified intensity was reported only in two studies [19,20] and differed between studies. Exercise duration was reported adequately in all studies. Options for types of aerobic exercise were provided and reported, but the type of exercise patients selected was not reported in any study. Intervention duration was reported adequately in all studies. It was not specified in any study whether group-based exercise was provided. In terms of supervision, all studies provided some form of supervision, either fully during the intervention period [19,20,26] or partly (1/3 of the program) [9,18]. It is unknown whether education on aerobic exercise prescription was provided before the start of the exercise program. It is important to note that only two studies clearly reported their methods to ensure adherence to the program [19,20]. This is especially important for studies that provided part or all of the program without supervision [9,18,25,26]. Only two studies clearly reported that were no adverse effects from the interventions [19,20]. No study reported on dietary intake or activity control during the intervention period. Furthermore, only two studies clearly reported on the inclusion of recovery strategies following exercise execution [9,18]. Lastly, prior exposure to aerobic exercise was not reported in any of the included studies.

## 4. Discussion

The purpose of this systematic review was to determine the effectiveness of aerobic exercise for patients with NNP and to identify the aerobic exercise variables that may have a positive effect on pain, function, and disability in this clinical population. Another aim of this review was to provide recommendations for future practice and research. It is important to note that to our knowledge, this is the first systematic review that examined the aforementioned intervention in this population and the first that examined the effectiveness of exercise variables related to intensity, frequency, duration, delivery, supervision, and adherence to the program. Overall, aerobic exercise was superior to other interventions or a control group; however, the program variables showed significant heterogeneity.

Based on our literature search, we found six eligible studies [9,18,19,20,25,26]. These studies used an exercise frequency of 2–3 times per week. However, based on previous research in healthy [29] and clinical populations [30,31,32], only two studies [18,26] met the published recommendations (at least three times per week). Furthermore, justification for the prescribed frequency was not available in any study. This is especially important since medical history provision and medical screening information for the participants would have probably supported and probably justified some of the prescribed exercise variables.

The intensity of the aerobic exercise was moderate to high in all studies. However, three studies [9,18,25] used subjective measures to assess the intensity and not objective measures such as maximum heartrate of heartrate reserve, as previously suggested [33]. Subjective measures of exercise intensity, such as the BORG scale, are less accurate [34]. Objective measures of exercise intensity are preferable in clinical trials since they make replication of future studies and protocols feasible and reduce the risk of exercise overestimation, especially in clinical populations with low exercise tolerance [35]. In three moderate-quality studies that used high-intensity exercise [9,18,26], medical screening was not reported in the methodology, making this program relatively unsafe, especially when it is not supervised [33] and in sedentary participants [36]. Again, justification for the provided exercise intensity was missing in all studies. Based on previously reported guidelines in healthy and clinical populations, for the provision of moderate (55–69% HRmax or RPE 12–13) or high exercise intensities (70–89% HRmax or RPE 14–16) in inactive individuals with or without comorbidities, medical screening and supervision should be considered since cardiac events are more likely to occur [32,36]. Additionally, a graded aerobic exercise program at a low to moderate intensity is preferable in patients with musculoskeletal pain [37]. This makes justification of the provided intensity much more controversial. Therefore, proper recommendation of the appropriate aerobic exercise intensity for patients with NNP is not possible from the available studies.

In terms of intervention duration, this ranged from 6 to 12 weeks, except for one moderate quality study that carried out the intervention for 12 months [26]. This is in line with previous reported studies in patients with non-specific low back pain [38]. Furthermore, most of the included studies provided long-term follow-ups except for one high-quality study [25], which enhanced the validity of the positive findings of aerobic exercise in NNP. However, justification of program duration was not available. This should be clearly justified in future studies.

It was not clear in the available studies whether group interventions of aerobic exercise were provided. Based on the information provided in the Methodology section of the included studies, it seemed that group interventions were not part of the interventions. Previous research in older adults has shown that both group interventions of aerobic exercise and individualized exercise can have significant health benefits; however, group exercise may be superior regardless of the total frequency of the exercise program [39]. Further, social connectedness with group exercise may increase adherence to the program [40]. This should not be misinterpreted as a one-size-fits-all model of exercise prescription and delivery. All programs should be prescribed individually, and differences should exist in all variables of the program for each patient. However, social interaction by exercising with others may increase the efficiency of the program.

Other exercise variables such as supervision or adherence were not reported in the majority of the eligible studies. Supervision is essential in patients, especially when they are prescribed high-intensity aerobic exercise, for safety [32,36]. Furthermore, proper supervision could confirm whether all exercise targets were met (i.e., intensity). It is important to note that lack of supervision may result in self-preferred exercise intensities, especially in high-intensity exercise programs [41]. Furthermore, adherence rates were not reported clearly in the available studies. Research in patients with mental health problems has shown that exercise undertaken at the patient’s preferred intensity may increase exercise adherence when compared with aerobic exercise programs with prescribed intensities [42]. Thus, any attempts to conclude whether different exercise intensities may influence exercise adherence in patients with NNP are not feasible.

### 4.1. Recommendations

Overall, based on the FITT principle, progressive or not progressive aerobic exercise in patients with NNP comprising either cycling, brisk walking, aerobics, stationary bike, treadmill running, circuit training, or swimming, with or without neck strengthening exercises and pain education or health promotion interventions (type), undertaken at moderate to high intensities (intensity), 2 to 3 times per week (frequency), with sessions lasting 30 to 45 min and with a minimum duration of 6 weeks (time), may be beneficial. The benefits of aerobic exercise were found in pain, function, and levels of disability.

When comparing the aforementioned exercise variables that were found in this review with those found in patients with low back pain [39], it seems that these are similar. However, it should be noted that the exercise variables that were used for patients with low back pain were not found to be clinically significant [39]. Furthermore, when comparing the findings of this study with the recommendations from other reviews in healthy and clinical populations, it seems that there is a significant overlap in the recommended exercise prescription parameters [29,32,33,43]. However, since the clinical significance is questioned with similar exercise parameters in low back pain patients, other exercise variables should be evaluated in future studies.

### 4.2. Limitations

The findings of this study showed that aerobic exercise is beneficial in NNP; however, this study did not include trials that recruited patients with whiplash-associated disorders or neuropathies caused by cervical spine pathologies. Thus, the generalizability of the findings in other clinical populations with neck pain is not feasible. Further, the effects of other types of aerobic exercise (i.e., dancing) or unstructured exercise (leisure time physical activity) were not evaluated. It is also important to note that limiting the inclusion criteria solely to English-language publications may have affected the overall conclusions since we may have missed studies published In other languages. Lastly, since a variety of outcome measures was included in the study and a relatively small number of studies exist that evaluated the effectiveness of aerobic exercise in NNP, a meta-analysis was not possible, which limits the possibility for strong conclusions to be made at this point.

## 5. Conclusions

Aerobic exercise may be an effective intervention for the management of NNP. The available evidence showed that aerobic exercise at a moderate or high intensity for 30–45 min for at least 6 weeks may result in beneficial outcomes. However, the available clinical trials are limited by poor reporting of their program variables and their justification for selection. Reporting standards should be implemented to allow for the replication of these studies in the future. More research is necessary in order to identify the minimum effective dose required for substantial health benefits in patients with NNP.

## Figures and Tables

**Figure 1 healthcare-11-00339-f001:**
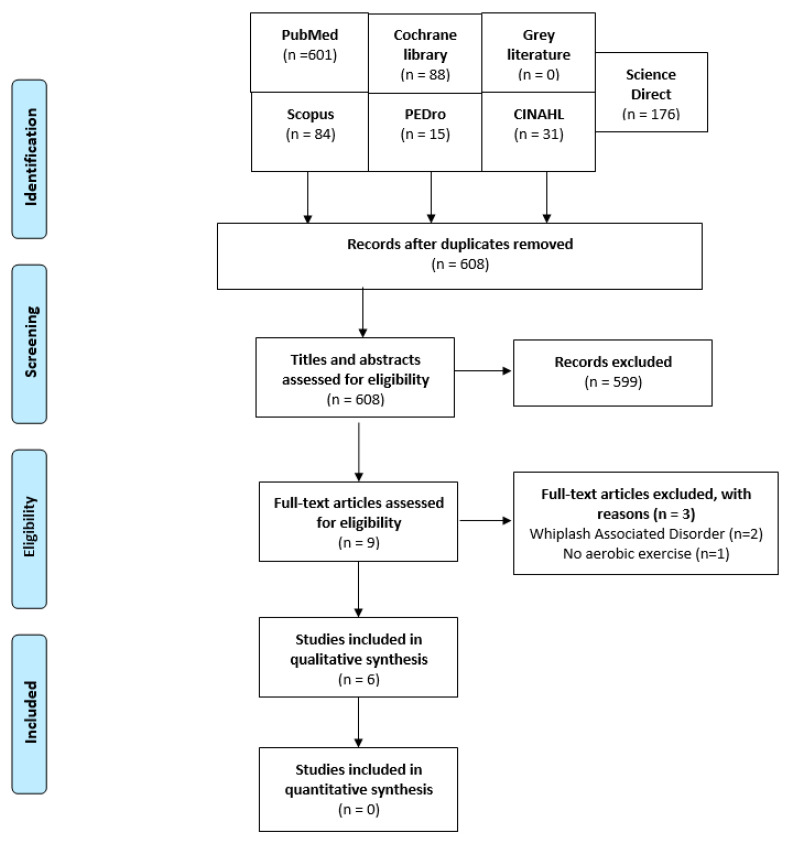
Preferred Reporting Items for Systematic Reviews and Meta-Analyses (PRISMA) flow diagram of the search results.

**Table 1 healthcare-11-00339-t001:** Characteristics of the included studies [9,18,19,20,25,26].

Studies	Characteristics of the Sample			Characteristics of the Intervention					Outcome Measures	Aerobic Exercise Intensity/Duration/ Type/Supervision
	Sample/ Diagnosis/Women/Study Design	Dropout	Mean Age ± SD (yo)	Interventions	Number of Sessions	Frequency (Times/wk)	Period (wk)	Long Term Follow-Up (Months)		
Daher et al., 2021	N = 139; ΝΝP; Female: 106 Randomized trial	0	(A) 55 ± 10.4 (B) 54.1 ± 10.7	**(A) Aerobic Exercise group:** neck-specific exercise PLUS moderate cycling exercise (60% of the age-predicted maximum heartrate) for 20 min during the first week, 30 min during the second week, and 45 min during the third to sixth weeks (N = 69)	12	2	6	3 and 6 months	Work Ability Index (WAI), Global Rating of Change (GROC)	Moderate (60% of the age-predicted maximum heartrate)/20 min during the first week, 30 min during the second week and 45 min during the third week and the remaining six weeks/cycling exercise/Physiotherapists supervised all exercise programs.
				**(B) Neck exercises:** supervised neck-specific exercise (N = 70)	12	2	6			
Daher et al., 2020	N = 139/NNP/Female: 106 (cont.) Randomized trial	17	(A) 55.0 ± 10.4 (B) 54.1 ± 10.7	**(A) Aerobic Exercise group:** cycling exercise (60% of the age-predicted maximum heartrate) for 20 min during the first week, 30 min during the second week and 45 min during the third week and the remaining six weeks PLUS supervised neck-specific exercise (N = 62)	12	2	6	3 and 6 months	Visual Analogue Scale (VAS), Neck Disability Index (NDI), Fear Avoidance Beliefs Questionnaire (FABQ) and cervicogenic headache	Moderate (60% of the age-predicted maximum heartrate)/20 min during the first week, 30 min during the second week and 45 min during the third week and the remaining six weeks/cycling exercise/physiotherapists supervised all exercise programs.
				**(B) Conventional group:** supervised neck-specific exercise (N = 60)	12	2	6			
Krøll et al., 2018 ^(a)^	N= 70/Migraine and co-existing tension-type headache and neck pain/Female: 62 Randomized trial (cont.)	18	(A) 42 ± 10.9 (B) 36 ± 10.1	**(A) Aerobic Exercise group:** 45 min of bike or cycling or brisk walking at a moderate intensity (14–16 RPE). (N = 26)	36	3	12	6 months	The number of days with TTH and NP, pain intensity, pain duration, area under the curve of duration * pain intensity for migraine, TTH and NP, physical fitness, level of physical activity, psychological well-being, and perceived ability to perform daily activities.	Moderate to high. The exercise period was divided into 10 min of warm-up (corresponding to 11–13 RPE), 30 min of endurance training (corresponding to 14–16 RPE), and 5 min of cool-down (corresponding to 11–13 RPE)/45 min/Bike or brisk walking/physiotherapist supervised 1/3 of the program.
				**(B) Control group:** did not receive any type of pain-modulating treatment. (N = 26)	-	-	12			
Krøll et al., 2018 ^(b)^	N = 70/Migraine and co-existing tension-type headache and neck pain/Female: 62 Randomized trial	18	(A) 42 ± 10.9 (B) 36 ± 10.1	**(A) Aerobic Exercise group:** 45 min of bike or cycling or brisk walking at a moderate intensity (14–16 RPE). (N = 26)	36	3	12	6 months	Pericranial tenderness, pain thresholds, supra-thresholds, and temporal summation	Moderate to high. The exercise period was divided into 10 min of warm-up (corresponding to 11–13 RPE), 30 min of endurance training (corresponding to 14–16 RPE), and 5 min of cool-down (corresponding to 11–13 RPE)/45 min/Bike or brisk walking/physiotherapist supervised 1/3 of the program
				**(B) Control group:** did not receive any type of pain-modulating treatment. (N = 26)	-	-	12			
Korshøj et al., 2017	N = 116; NNP; Female: 88 (cont.)	11	(A) 44.9 ± 9.2 (B) 45.7 ± 8.1	**(A) Aerobic exercise group:** Aerobic exercise ≥60% maximal oxygen consumption VO2 max with the following: aerobics, biking on a stationary bike, treadmill running, and circuit training for 30 min. (N = 57)	40	2	16 and 48	4 and 12 months	Standardized Nordic Questionnaire for the Analyses of Musculoskeletal Symptoms for the following areas: neck, shoulders, arms/wrists, upper back, lower back, hip, knees, and feet/ankles.	Moderate. Heartrate reserve of 67%/30 min/aerobics, biking on a stationary bike, treadmill running, and circuit training/supervised (not reporting who).
				**(B) Reference group:** lectures in health promotion (N = 59)	2	2 h	-			
Brage et al., 2015	N = 15; NNP; Female: 15 Randomized trial	0	(A) 42.14 ± 10.8 (B) 40.7 ± 13.6	**(A) Pain education and exercise group:** pain education (90 min) and specific training (neck-shoulder exercises, balance and aerobic training). Aerobic exercise included walking, jogging, swimming, cycling (N = 7)	4 Pain education 8 for exercise	-	8	No follow-up	Neck pain, function and Global Perceived Effect (GPE), Surface electromyography (EMG) from neck flexor and extensor muscles during performance of the Cranio-Cervical Flexion Test (CCFT) and three postural control tests (two-legged: eyes open and closed, one-legged: eyes open)	Moderate; 11 and 14 on a Borg scale/The starting duration was set to 20% below the patient indication and progressed weekly by increasing the duration of training by 20% (up to a maximum of 30 min)/walking, swimming, cycling, jogging, or stick walking/no supervision
				**(B) Pain education group:** Pain education only (90 min) (N = 8)	4 Pain education		8			

Mo, months; NA, not available; SD, standard deviation; VAS, Visual Analog Scale; TTH, tension-type headache; NP, neck pain; Wk, week; ROM, range of motion; RPE, rate of perceived exertion; NNP, non-specific neck pain; FABQ, Fear-Avoidance Beliefs Questionnaire.

**Table 2 healthcare-11-00339-t002:** Methodological quality assessment using the PEDro scale [9,18,19,20,25].

Studies	1	2	3	4	5	6	7	8	9	10	11	Total Score
Daher et al., (2021)	Y	Y	Y	Y	N	N	N	Y	Y	Y	Y	7
2. Daher et al., (2020)	Y	Y	Y	Y	N	N	Y	Y	Y	Y	Y	8
3. Krøll et al., (2018)a	Y	Y	Y	Y	N	N	N	N	N	Y	Y	5
4. Krøll et al., (2018)b	Y	Y	Y	Y	N	N	N	N	N	Y	Y	5
5. Korshøj et al., (2018)	Y	Y	N	Y	N	N	N	N	Y	Y	Y	5
6. Brage et al., (2015)	Y	Y	Y	Y	N	N	Y	N	Y	Y	Y	7

Y: Yes; N: No. 1. eligibility criteria; 2. random allocation; 3. concealed allocation; 4. baseline comparability; 5. blinding of individuals; 6. blinding of therapists; 7. blinding of assessors; 8. adequate follow-up; 9. intention-to-treat analysis; 10. between-group comparisons; 11. point estimates and variability. Item 1 (eligibility criteria) does not contribute to the total score.

## Data Availability

Not applicable.

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
