# Peer review of "A Systematic Review of the Aerobic Exercise Program Variables for Patients with Non-Specific Neck Pain: Effectiveness and Clinical Applications"

_healthcare, 2023, doi:10.3390/healthcare11030339_

Round 1

Reviewer 1 Report

The paper is very interesting and it discuss a very important topic concerning the relationship between aerobic exercise and pain. The methods are clear and methodology is respected.

The topic is particularly interesting expecially for its impact on healthcare system, quality of life of of patients. 
The paper is suitable for publication I I coud just suggest some implementation for the background session:
- consider the impact of physical activity on physical and psychical outcome (10.1016/j.archger.2020.104109 and 10.1016/j.jamda.2019.01.128)
- consider the impact of polymedication on older and mood (10.1017/S1041610217001715 and 10.1007/s40520-018-0893-1)
Thanks

Reviewer 2 Report

Line 39:  Studies have shown that exercise remains the mainstay treatment for the management of NNP (7). Give more citations if you use the plural form.
Line 46: myofascial pain(11), and pain perception(12). Add space before brackets.
Figure 1. In the box add a number of excluded studies and the reasons.

Line 156: from 36 to 55 years old. Also,
Line 170: was 30(21, 22) or 45(14-17): Add space before brackets.
Chapter 3.3.- Add space before brackets.

Reviewer 3 Report

The work done by the authors contributes significantly to the field of research.

I recommend that authors try to amend their work according to the below suggestions.

What was the difference between the systematic review that was conducted by O'Riordan et al. 2014 (reference 13) and your review? Is it only updating the review? Please add the main differences in brief.

Since authors are discussing aerobic exercise, they should add about FITT (each variable individually) principle in the introduction and link it and discuss it in the discussion.

There are differences between the methods on the registered protocol and the actual search happened. Have you searched ScienceDirect, and Cochrane Library? Please update.

Instate of a shorter period, please provide elaboration or justification to have the database from inception to 2022 (when? Is it 1900? When is the inception in your case?)

To clarify the scene for the reader, I recommend that reporting the results and therefore the conclusion be based on the quality of the included trials. For example, 2 high quality trials (X,Y) reported beneficial effect of AE .. etc.

To assists with the above comment, check Cochrane risk of bias2 (CROB2) tool, which apply strict measures compared to PEDro, especially that the included studies are RCTs.

Reviewer 4 Report

Thank you for the opportunity to review this interesting paper about the effectiveness of aerobic exercise programs in patients with non-specific neck pain. I congratulate you on the methodological rigor of the review. I make some comments about the work and mention some aspects that should be reviewed.

INTRODUCTION is very brief, although it allows to situate and justify the study in a convenient way.

MATERIAL AND METHODS

Study selection: Limiting the search by language is a restrictive criterion because not all published documents on the subject are retrieved. It is better not to use this filter, so that the search is as exhaustive as possible. In this case, the Limitations of the study could reflect that documents of interest written in other languages may be missing.

RESULTS

The first paragraph (lines 143-147) are not results, but are part of the search strategy and must be reflected in the Methodology.

PRISMA flow diagram has some error. If the Records after duplicates removed are n=601, in the following section they cannot be n=608.

Table 1: layout it correctly. If it appears cut off, add “(continuation)” to the heading. In the legend of Table 1 (lines 163,164) add the development of the abbreviations NNP and FABQ.

REFERENCES

- According to Vancouver style rules, journal titles should appear abbreviated.

- It is recommended to add DOI to the articles that have it.

- References No. 20, 29: missing page data.

- References no. 33 and 34 are repeated. Delete one and review the citations in the text.
